Who participates in computer science education studies? A literature review on K-12 subjects

van der Meulen Anna a.n.van.der.meulen@liacs.leidenuniv.nl 1
Hermans Felienne 1
Aivaloglou Efthimia 1 2
Aldewereld Marlies 1
Heemskerk Bart 1
Smit Marileen 1
Swidan Alaaeddin 2
Thepass Charlotte 1
de Wit Shirley 1 3
1 Leiden Institute of Advanced Computer Science, Leiden University , Leiden , The Netherlands
2 Open University of the Netherlands , Heerlen , The Netherlands
3 VHTO , Amsterdam , The Netherlands
Piccolo Stephen
Electronic publication date: 2021 Dec 21
Publication date: 2021
Volume: 7
Electronic Location ID: e807
Received 2021 Jul 20; Accepted 2021 Nov 16
Copyright: ©2021 van der Meulen et al.
Copyright year: 2021
Copyright holder: van der Meulen et al.
License: This is an open access article distributed under the terms of the Creative Commons Attribution License, which permits unrestricted use, distribution, reproduction and adaptation in any medium and for any purpose provided that it is properly attributed. For attribution, the original author(s), title, publication source (PeerJ Computer Science) and either DOI or URL of the article must be cited.
License URL: https://creativecommons.org/licenses/by/4.0/

Keywords: Computer science education, K12, Literature review

Funding: The authors received no funding for this work.

==============================
Computer science education (CSEd) research within K-12 makes extensive use of empirical studies in which children participate. Insight in the demographics of these children is important for the purpose of understanding the representativeness of the populations included. This literature review studies the demographics of subjects included in K-12 CSEd studies. We have manually inspected the proceedings of three of the main international CSEd conferences: SIGCSE, ITiCSE and ICER, of five years (2014–2018), and selected all papers pertaining to K-12 CSEd experiments. This led to a sample of 134 papers describing 143 studies. We manually read these papers to determine the demographic information that was reported on, investigating the following categories: age/grade, gender, race/ethnic background, location, prior computer science experience, socio-economic status (SES), and disability. Our findings show that children from the United States, boys and children without computer science experience are included most frequently. Race and SES are frequently not reported on, and for race as well as for disabilities there appears a tendency to report these categories only when they deviate from the majority. Further, for several demographic categories different criteria are used to determine them. Finally, most studies take place within schools. These insights can be valuable to correctly interpret current knowledge from K-12 CSEd research, and furthermore can be helpful in developing standards for consistent collection and reporting of demographic information in this community.

Introduction

Computer Science Education (CSEd) is being increasingly taught in schools, starting as early as kindergarten. As a result, CSEd research, aimed at children under the age of 18, has also been on the rise. While several papers study teachers, programming environments, or curricula, many CSEd papers observe children as participants, using a variety of research methods from case studies to controlled experiments. These studies might have a large impact on policy, since many countries are currently in the process of implementing mandatory programming and computer science curricula (Barendsen, Grgurina & Tolboom, 2016; Hong, Wang & Moghadam, 2016). Further, reporting complete data on the participants and the settings of K12 CSEd activities is also required for activity comparison and replication across studies. Previous work focusing on CSEd studies and their reporting of activity components, including student demographic information, found that race and socio-economic (SES) status are rarely reported (McGill, Decker & Abbott, 2018). We build upon this, by focusing specifically on participant demographic data, including information which has previously not been analyzed, such as the presence of students’ disabilities. Although data on disabilities is of an especially sensitive nature and it might not always be possible or appropriate to obtain this information, we believe it is both relevant and important to consider as a demographic factor. An estimated 15% of the world population has a form of disability, (https://www.un.org/development/desa/disabilities/resources/factsheet-on-persons-with-disabilities.html) and inclusivity in CSEd can be helped forward if it is better understood how specific subpopulations of children can, when needed, be additionally supported. Further, we examine in depth the types of data that are reported or omitted, by assessing who is reported on (for instance, how many boys and girls for the category gender) and how a category is reported on (for instance, which indicators are used to represent SES).

We are interested in gaining an understanding of the representativeness of the children who are participating in K-12 CSEd studies. To that end, we have conducted a two-phase literature review of the proceedings of five years (2014–2018) of the SIGCSE, ITiCSE and ICER conferences. In Phase 1, we manually inspected the abstracts of all papers of at least 6 pages in length (953 papers) and then identified all papers that (A) concerned K-12 (pre-university) subjects and (B) involved subjects taking part in CSEd activities. 134 papers remained. Then in Phase 2, we read all the full papers, and gathered the reported demographic information of the participants in the categories: age/grade, gender, race/ethnicity, location, prior experience, SES and disabilities.

The aim of this literature review is to gain insight in the demographics of subjects participating in K-12 CSEd studies. These insights can contribute to first of all understanding the representativeness of the population included in these studies at the moment, thus correctly informing policy and enabling comparisons and replications. Second, our paper can help researchers identify which characteristics are important to report on when conducting CSEd studies.

K-12 computer science education

CSEd, and the broader computational thinking (Wing, 2006), has recently been made part of the curriculum in many countries (Barendsen, Grgurina & Tolboom, 2016; Hong, Wang & Moghadam, 2016). Especially block-based programming languages are currently being used in K-12 programming education extensively, most commonly using Scratch (Resnick et al., 2009) and Alice (Conway et al., 1994). In addition to blocks-based languages, robots (Kazakoff, Sullivan & Bers, 2013; Ludi, Bernstein & Mutch-Jones, 2018; Swidan & Hermans, 2017) and textual programming languages (Hermans, 2020; Price & Barnes, 2015; Swidan & Hermans, 2019) are also frequently used. Other approaches use no computers at all, referred to as unplugged computing (Hermans & Aivaloglou, 2017).

A SIGCSE paper (Al-Zubidy et al., 2016) showed that empirical validation is common among SIGCSE papers, with about 70% of papers using some form of empiricism. There is a large variety in methods however: studies are performed both in the classroom (Grover, Rutstein & Snow, 2016; Hickmott, Prieto-Rodriguez & Holmes, 2017; Kazakoff & Bers, 2012) as well as in extracurricular settings (Maloney et al., 2008). Not all studies involving K-12 CSEd include subjects: in addition to small scale studies in classrooms, programs created by learners have also been analyzed by researchers. Scratch projects were used for example to explore the learning patterns of programmers in their first 50 projects (Yang et al., 2015). Aivaloglou & Hermans (2016) performed an analysis on 250.000 Scratch programs investigating the occurrence of code smells.

Research on demographics in computer science

There is some previous research aimed at understanding who participates in K-12 computing. Ericson and Guzdial, for example, in their exploration of the CS Advanced Placement test, specifically investigate women and non-white students (Ericson & Guzdial, 2014). Some papers also specifically looked at the reporting of demographics of students in computer science papers. In a meta-analysis from 2018 (McGill, Decker & Abbott, 2018) 92 articles from SIGCSE, ICER, and ToCE between 2012 and 2016 were studied, analyzing which components or elements of activities are reported in K-12 studies, including data on the activities, the learning objectives, the instructors and the participants. Regarding participant demographics, they found that, while age and grade of participants are often reported on (98% and 74%), other factors are reported less, such as gender (64%), race (45%) and SES (13%). A similar study on CHI papers between 1982–2016 (Schlesinger, Edwards & Grinter, 2017) found that gender is reported in 63% of papers, race in 13% and class in 23%. The reporting of combined demographics is even more rare; only 2% of papers in the field of HCI report gender, race and class of subjects.

Our study expands these prior, related, works in two ways. First, our focus is on the content of the demographics, studying in depth who participates in CSEd studies, in addition to the information on whether the demographics are being reported on. Second, we performed a full manual analysis of an expanded set of 134 papers of the three main international conferences within the CSEd community from 2014 to 2018, in order to also gain insights in the way in which demographic information is presented and possible reasons why this information is omitted. This includes the factor disabilities, which previous reviews (McGill, Decker & Abbott, 2018; Schlesinger, Edwards & Grinter, 2017) did not address as a demographic factor. A high prevalence of individuals with disabilities exists worldwide (https://www.un.org/development/desa/disabilities/resources/factsheet-on-persons-with-disabilities.html), yet this group remain underrepresented in fields of computing and software engineering (Burgstahler & Ladner, 2006), and accessible technology remains lacking (Patel et al., 2020). Attention for these issues in CSEd research is rising, and several research lines on young learners with disabilities have started, often with focused questions and target groups (Hadwen-Bennett, Sentance & Morrison, 2018; Israel et al., 2020; Powell et al., 2004). Insights on support these learners need to be fully included in CSEd is growing, and can concern adapted or assistive technologies as well as instructional approaches. Especially since the majority of children with disabilities takes place within classes in regular education (Van Mieghem et al., 2020), it is important to assess their representativeness within CSEd studies to further understand how inclusive CSEd education can be fostered. Together, these insights are valuable to understand the representativeness of the populations included in K-12 CSEd studies, to interpret previous and future findings, and to help guide reporting on future work.

Finally, one point to address concerns our choice of the three conferences. We believe that the proceedings of the SIGCSE, ITiCSE and ICER conferences reflect the global CSEd community well as they are often referred to as the “main international conferences”, and consequently provide a valid starting point to explore the demographics of its studies subjects. It is likely that the US will be over-represented in our findings, however, it can be considered part of our assessment to determine this and to reflect on the implications. A related point concerns the demographic category race or ethnicity. This category can be common to include, as the previous studies on demographics of K-12 computing participants confirm (McGill, Decker & Abbott, 2018; Schlesinger, Edwards & Grinter, 2017), however, this strongly depends on the context and country. We will follow the terminology used in the studies we include, taking this together under the term “race/ethnic background”. In addition, we will also explore whether the inclusion of the category of disabilities is related to context and country.

Survey Methodology

Setup of literature review

The research question of this paper is: What are the demographics of subjects who participate in CSEd studies in K-12? In order to answer this question, a two-phase literature review was conducted. In Phase 1, all 953 papers that appeared in the proceedings of one of the three main international CSEd conferences: SIGCSE, ITiCSE and ICER between 2014 and 2018, were inspected using the procedure and criteria described below to determine whether they qualified for Phase 2. The resulting 134 papers were manually analyzed to collect the reported demographic information of each paper. The distribution of papers in each phase is shown in Table 1.

Table 1 Number of papers analyzes in this paper per conference.

Conference	Phase 1	Phase 2	
ICER	125	17	
ITiCSE	267	25	
SIGCSE	561	90	

Paper inspection (phase 1) and paper analysis (phase 2)

In Phase 1 all full papers (six pages or longer) which appeared in the proceedings of SIGCSE, ITiCSE and ICER in five years (2014–2018) were inspected. Phase 1 included 953 papers in total (561, 267 and 125 respectively for the three conferences). These papers were inspected by reading their title and abstract. The first papers were inspected together by the group of nine authors to evaluate the inspection process, after which the rest of the papers were divided among all authors. Papers were inspected individually, but uncertainties concerning whether a paper qualified for Phase 2 were noted down and discussed among the group of authors together until agreement was reached. For papers to be included in Phase 2, we set two criteria:

K-12 Participants in the reported study should be K-12 level: in kindergarten, elementary school, middle school or high school. Papers that described studies of which some of the participants were K-12 while others were already in university were also considered.

Computer Science Education activity Participants in the reported study should be actively involved in CSEd activities. We excluded studies in which children did not actively engage in programming or other CSEd activities, for example, studies in which only children’s attitude towards programming or computers was measured without any intervention.

In Phase 2 the papers selected during the first phase were analyzed. This concerned 134 papers (13% of all SIGCSE, ITiCSE and ICER 2014 to 2018 papers). Each paper was carefully read in full to gather the demographic information that was reported on or to note the lack of report of information. The specific demographic categories are described in the next section. The same process as in Phase 1 was followed, where the first three papers were read together by all authors to practice the process. Next, the papers were divided among the nine authors. Uncertainties were noted down and discussed by the authors in a weekly group meeting. In these meetings, demographic information that was inferred and not explicitly stated in the papers, such as information on location, was checked and agreed upon by the authors. Moreover, coding conventions were set for demographic information that was not uniformly reported on, such as age and school system levels, race/ethnicity and socio-economic status.

Demographic categories

The demographic categories were determined at the start of Phase 2, observing the categories commonly reported in the included papers. We confirmed the demographic categories that McGill found (McGill, Decker & Abbott, 2018): ages/grades, gender, race/ethnic background, location, prior experience and SES, and included the factor disabilities.

Results

This section presents the results of our analysis of the 134 papers reporting on CSEd studies using K-12 students as subjects. We found 9 papers in the sample that reported on multiple studies, and we decided to classify them separately, because they occassionally reported on different samples. As such in the remainder of the paper, we report on the number of studies, not the number of papers. The 134 papers in total report on 143 studies. Table 2 presents an overview of the number of studies reporting on the different demographic categories. Not all conferences were represented equally, as can be seen in Table 1. This is mainly due to the fact that SIGCSE is bigger in terms of papers than the other two conferences. Our sample represents the conferences quite equally: 16% of SIGCSE papers were included, 9% of ITiCSE papers and 14% of ICER papers.

Table 2 Number of studies (out of 143) that report on the different demographic categories of K-12 participants.

Demographic category	Number of studies	Percentage	
Age/grade	134	94%	
Gender	92	64%	
Race/ethnic background	46	32%	
Socio-economic status	28	20%	
Disabilities	6	4%	
Location	96	67%	
Prior experience	52	37%	

Before we present the results of the demographic categories, we briefly refer to the context in which the studies are performed. All but two studies report on the context where the experiment was run. The majority of studies (88 or 62%) took place within a school, while 49 (34%) took place outside of school, for example in a coding club or summer camp. Four studies report performing experiments or interventions both in a school and in another context.

Age and grade

The ages of the participants, or their grade, is reported in almost all papers in our sample: 134 studies (94%) report this information. Since the majority of papers refer to the classification of the US school system of elementary, middle, and high-school, we followed this classification as well in order to obtain an overview. Information from papers that reported on a different grade system or only on age was manually classified. We classified Kindergarten to grade 6 as elementary school, grades 6 to 8 as middle school and grades 9 to 12 as high school. Grade 6 is within the US classified as either elementary or middle school, depending on the specific school. Consequently, where school was provided in a study for grade 6 we follow this and noted either elementary or middle school. Where school level was not provided, we classified grade 6 as elementary school unless it was indicated as part of a range that includes higher grades a well (for instance, participants being from grades 6-8), in which case we classified the whole range as middle school. Middle school and high school participants are most common in the papers, representing 37% and 31% of studies respectively. Some papers report on a variety of subjects in their studies, for example one paper reported on a study with participants from pre-school to university (Martin, Hughes & Richards, 2017).

Gender

Gender is relatively well-reported, with 92 studies (64%) providing the gender of their participants. As Fig. 1 shows, 60 studies (44 + 16), which together represent 65% of studies that report on gender, have over 55% male participants. Even though boys are over-represented in general, girls are over-represented in studies focusing specifically on gender: out of the 12 studies with less than 25% male participants, six consisted exclusively of female students. Further, the majority (10) of the studies with less than 25% male participants took place in camps or after school. In the studies with less than 25% female participants this pattern was not visible. Besides the studies that included just one gender, there are also studies that included gender in their research purpose (22% of the studies that reported gender). Overall, male participants appear over-represented in the studies which report gender, even though there are several papers with a special-focus on female participants. Finally, two studies report in addition to male and female the category other, which includes 1% (Paspallis et al., 2018) and 5% (Schanzer, Krishnamurthi & Fisler, 2018).

Figure 1 Studies categorized by the reported percentage of male students.

Race/ethnic background

With 46 studies (34%) reporting on the race or ethnic background of their participants, this category is reported on in approximately half as many studies as gender. Out of these 46, 37 studies provide concrete quantifiable information. The remaining studies for instance indicate race about a proportion of their participants but leave a large proportion as “unknown” (Wang & Hejazi Moghadam, 2017) or refer merely to different countries (Srikant & Aggarwal, 2017). The quantifiable information from the 37 studies is represented in Fig. 2, showing that white students form the minority of participants, with 13 studies (28%) reporting 0 to 25% white students. Figure 3 depicts the number of races or ethnicity’s included in the studies that provide this information.

Figure 2 Studies categorized by the reported percentage of white students.

Figure 3 Studies categorized by the reported number of different races/ethnicities.

As is the case for gender observations, there are also a number of papers reporting on interventions, tools or programs specifically aimed at ethnic minority students. Out of the 46 studies that reported race/ethnicity, 15 (or 33%) studies included race/ethnicity in their research purpose. Some of these studies that target a specific race/ethnic group also target a specific gender, focusing on African-American girls (Thomas, 2018; Van Wart, Vakil & Parikh, 2014), or American-Indian boys (Searle & Kafai, 2015).

Socio-economic status (SES)

SES is reported in 28 studies (20% of studies). Figure 4 gives an overview of what is being reported in these studies. The figure shows that there is no clear focus on either low or high SES students, and that many papers explicitly stated the SES of subjects was mixed. Out of the 28 studies that report on SES, 8 (29%) included SES within their research purpose.

Figure 4 Studies categorized by the reported percentage of students with low socio-economic status (28 studies in total).

We also found that SES was approached differently across the studies. Various indicators were used, including income or poverty rate, as well as eligibility of the sample or the population for free lunches or other forms of extra funding, which all represent slightly different aspects of SES.

Disabilities

With only six studies (4%) reporting on disabilities of their participants, this category is the least reported on. Out of the six, two studies named visually impaired children (Kane & Bigham, 2014; Ludi, Bernstein & Mutch-Jones, 2018), and one named children with autism (Johnson, 2017). In one of the studies that included children with visual impairments (Kane & Bigham, 2014) and in the study that mentioned children with autism (Johnson, 2017), these target groups are part of the context in which the research takes place (a camp for students with autism and a camp for visually impaired children, resp.). Consequently, the information on the prevalence of these disabilities seems inferred from the research context and not explicitly acquired. In addition, no other details (such as specific types within these syndromes) are provided. The other study on children with visual impairments (Ludi, Bernstein & Mutch-Jones, 2018) provides somewhat more information, indicating that the participants were recruited across the US, from various educational situations (homeschooled, public school, schools for the blinds), and furthermore that 58% had low vision and the remaining children were blind.

The other three papers, out of the 6 that reported on disabilities, remarked at a more general level that “students with various disabilities” participated in their study (Paramasivam et al., 2017) or that there were children with “special needs or learning difficulties” (Grover, Rutstein & Snow, 2016) or from “special education” (Grover, Pea & Cooper, 2016). The paper by Paramasivam et al. (2017) provided a wide range of disabilities included in their sample (“deafness, low vision or blindness, Cerebral Palsy, Muscular Dystrophy, Ollier’s disease, Attention Deficit Disorder, Asperger’s Syndrome, and other autism spectrum disorders or learning disabilities”), without however specifying which disability was present for how many students. All six studies that report on disabilities concern research conducted in the US.

Two additional papers reported comparable cognitive background information on their participants by describing that the children in the study were gifted (Friend et al., 2018), or were students in a school which is also a center for gifted children (Daily et al., 2014) (which is usually not seen as a disability, consequently we did not include these 2 papers in the 6 papers of this category).

Location

Location is a multi-faceted factor, which (McGill, Decker & Abbott, 2018) describes as “Specific locations, including city, state, and country”. Country is reported on relatively often, with 96 studies (67%) explicitly stating the country in which the study is being executed. For 42 studies (29%), the location of the study can be deduced with reasonable certainty from the description of the setup in the paper, for example when the authors work at a university in the US and state they have executed the study themselves.

Combining papers that explicitly report subjects from the United States (65) and papers where we deduced the US as location from context (36), we find 101 studies (71% of all studies) in which children from the US participate. This overshadows studies conducted in Europe (19 or 13%) and Asia & Australia (seven or 5%). Location more specific than the country of the study is reported on in about one in three papers (46 studies). When it is reported, however, it is often with different terms focused on different aspects. For example, some papers state the location along with more information about the residents, such as “two tribes of an American Indian community outside of Phoenix, Arizona” (Kafai et al., 2014), while others are more generic and use broad locations such as “Western Germany” (Pasternak, 2016) or “a large Northeast city” (Deitrick et al., 2015).

Prior experience

Prior experience with an aspect of computer science is reported in 52 studies (37%). However, it is typically reported on loosely using phrases such as “some children had coding experience” (Broll et al., 2017), “students did not begin the class with strong experience in computers and computing” (Freeman et al., 2014) or “most had little to no experience with robotics design or programming.” (Ludi, Bernstein & Mutch-Jones, 2018) which leaves room for interpretation as to what some, strong or most means exactly. Figure 5 shows the distribution of the studies over different categories of the experience of subjects. The category “mixed” pertains to studies for which it was explicitly reported that some of the participants have experience while others do not. As can be seen in Fig. 5, the largest category of studies (22, 42% of studies reporting on prior experience) reports on subjects without any experience. In some cases, papers reported on excluding children with prior knowledge of computer science through a pretest or self-assessment, for example (Zhi, Lytle & Price, 2018).

Figure 5 Studies categorized by the reported percentage of students with experience in programming (52 studies in total).

Conclusions

With this literature review we aimed to gain insight in the demographics of subjects participating in K-12 CSEd studies, in order to understand the representativeness of the populations included, and in order to identify which characteristics empirical studies can report on. Our insights are discussed below.

Main overview of reported information and included populations

First, the reported demographic information shows that children from the United States, boys and children without experience appear over-represented. Second, we found that demographic information in general and specific categories is often missing. Especially race or ethnic background, disabilities and SES of participants are frequently not reported on, as previous studies also showed (McGill, Decker & Abbott, 2018; Schlesinger, Edwards & Grinter, 2017). Finally, most studies take place within schools. These insights are further discussed below, together with other observations on the inclusion and report of demographic factors.

US focus

As we expected, the category location showed that children from the US were over-represented in the studies in SIGCSE, ITiCSE and ICER we reviewed. This limits the representativeness of the findings of these studies first of all because of curriculum differences world-wide in the organization of K-12 education. In addition, in many European countries (including Sweden, Germany, Finland and the Netherlands) there are no or very few private schools, and homeschooling is in European countries often prohibited (in Greece, Germany and the Netherlands) or hardly practiced (in France). These differences can make it difficult to generalize findings. A second implication of the over-representativeness of the US is that the reporting is also distinctly US flavored. For example, when discussing SES, a paper stated that a school has “eligibility for Title I funding” (Ibe et al., 2018) or that children received “free or reduced lunch” (Buffum et al., 2016; Hansen et al., 2017) referring to distinct US policies that people outside the US might not be familiar with. Similarly, a concept such as an “urban” area (Grover, Rutstein & Snow, 2016; Tsan et al., 2018) does not have the same connotation in all countries; it can refer to a school in a poor inner city area or a well-educated high-income neighborhood.

Interpretation and generalization could be strengthened if empirical studies (including those from the US) give a clear report on factors of location and SES. This can be done either in way that is easy to understand without an in-depth knowledge of a specific school system and context, or by providing a brief description of the educational system in a country. Some of the included papers provide examples of this approach, for example when describing the specific setting in France (Chiprianov & Gallon, 2016). Further, it is important to be aware of the high presence of children from the US in studies in SIGCSE, ITiCSE and ICER, and to interpret findings with this in mind or explicitly look for studies from other locations.

Inconsistent criteria and school-level reporting

We observed that, for several demographic categories, different criteria were being used by different papers. This was most visible for SES, where some papers reported low SES based on eligibility for lunch programs (Buffum et al., 2016) while others used poverty rates (Feaster et al., 2014) or the economic and educational level of parents (Ko et al., 2018). The same is true for prior experience. Some papers excluded children with too much prior knowledge (Zhi, Lytle & Price, 2018), some papers named experience with the tool or language being used (Joentausta & Hellas, 2018) while others explicitly separated experience with the tool from computer science experience in general (Smith, Sutcliffe & Sandvik, 2014). To increase the clarity of future studies, it would be beneficial for our community to establish recommended guidelines for the criteria to be used for SES or prior experience in K-12 studies.

Further, both in the case of race/ethnic background and disabilities it can be considered whether a bias is occurring in reporting this information. First, in the case of race/ethnic background, the studies that provide quantifiable information on this demographic together suggest that white students form the minority, which is not in line with the American population (to which a large percentages of our studies pertain), where about 50% of children enrolled in public elementary and secondary schools are white (https://nces.ed.gov/programs/raceindicators/indicator_rbb.asp). It is possible that race or ethnic background is more often reported when participants do not belong to the racial/ethnic majority, or, specific to the current research setting, when their background is not congruent with expectations of computer science classroom populations. This is further confirmed by the large number of studies that report on mixed-racial experimental groups. Here as well, a higher number of mixed-racial groups is reported compared to what can be expected from typical classrooms in the US (Stroub & Richards, 2013). Second, a similar type of reporting bias might be occurring for the factor disabilities, which are mainly being reported when the study targets or focuses on children with disabilities and as such all children in the study undoubtedly have a disability (such as in a camp for blind children). Here as well, it seems unlikely that only 6 studies reporting on disabilities is an accurate representation of the population, where worldwide approximately 15% has a disability (https://www.un.org/development/desa/disabilities/resources/factsheet-on-persons-with-disabilities.html). However, for both of these factors (race/ethnic background and disabilities) the lack of consistent reporting might also be related to the especially sensitive nature of these factors, which is further discussed below.

Finally, some papers reported the demographics of the study not at the level of the study participants, but at the level of the school. For example: “At the elementary school where we collected the data, the student body is roughly 53% African-American, 33% Caucasian, and 14% Hispanic, Latino, Native American, Asian, or mixed race. Approximately 47.4% of the students receive free or reduced cost lunch” (Buffum et al., 2016). This presents only an indication of the exact study sample, which might differ because of factors such as selection bias, with some students or teachers being more willing or available to participate. At the same time, school level reporting is preferable over no reporting at all. Further, as discussed in the section below on ethical and legal considerations, it can be a good alternative when dealing with practical or ethical constraints.

Disabilities as a demographic factor

In this review we explored the inclusion of “disabilities’ as a demographic factor, which was not included in previous reviews on CSEd research populations (McGill, Decker & Abbott, 2018; Schlesinger, Edwards & Grinter, 2017). Only 6 studies reported on the presence of disabilities, referring for instance in a relatively general manner to children with visual impairments or to learners with special needs. Usually how this information was obtained was not indicated, though in some cases this was inferred from the targeted research context (such as a camp for children with autism). It is likely that the low and relatively general reporting on disabilities can be explained by a lack of acknowledgment of this factor as a standard demographic (as the absence of mentioning it in the previous reviews also suggests), and, related, the sensitivity of these data (Fernandez et al., 2016). The need to increase inclusivity of CSEd, especially concerning learners with impairments (Burgstahler & Ladner, 2006), stresses the importance of considering how a balance can be found in reporting on this information in a more consistent, but also feasible, way. A starting point might be to consistently refer to the consideration of the presence of disabilities within a sample, adding information where possible at an appropriate level of detail.

Further assessment of how studies outside of the conferences of our focus handle and report on the presence of disabilities can be helpful. Although a thorough exploration falls outside of the scope of this review, some suggestions can be found in other CSEd journals or conferences within the topic of learners with visual impairments. Similar to the paper by Ludi, Bernstein & Mutch-Jones (2018) included in our review, some of the often small scale studies within this topic tend to elaborate both on the specific type of impairment (including for example being blind from birth, or partially sighted) and on the recruitment of the participants (through special education schools or personal contacts) (Jašková & Kaliaková, 2014; Kabátová et al., 2012; Milne & Ladner, 2018; Morrison et al., 2018). An interesting option that is being applied is to elaborate on a participants’ disability in terms of use of computer, assistive technologies, or tools (Jašková & Kaliaková, 2014; Milne & Ladner, 2018). Although such level of detail can be highly useful to interpret the findings, it is likely that, in studies taking place in larger groups and within mainstream education, obtaining this information is not always feasible or practical. In that case, more general (indicating the presence of disabilities) or school level (indicating a percentage) report can be valid alternative options, that still provide an indication of the population included. Authors could then also explain or substantiate why they report in this manner. For instance, Koushik & Kane (2019) indicate in their study on training computer science concepts that their participants have cognitive disabilities. They further describe: “We did not collect individual diagnoses from our participants as we did not believe this personal information was relevant to our research goals”, and provide a summary of different types of cognitive impairments included in the club of which the participants were members. Overall, a tailored approach might be needed for the report on disability, yet mentioning the factor and substantiating the information that is (not) provided can be very helpful for the field.

Separate subjects section

While examining the papers, we found that papers do not only differ in which demographic data were reported, but also in how this was presented in the paper. 53 papers (40% of 134 papers) have a dedicated section or subsection where all information on participants was placed, called “subjects”, “participants”, or something similar. In 35 papers (26%), information about participants was placed in larger sections, such as “experimental design”, “research setup” or “methods”. In the remaining 38% of the papers there was no central place in the paper for demographic information. Instead, it was placed across different sections, or was only stated in the introduction or abstract. We would advise journals and conferences that are considering to adopt guidelines for the disclosure of participant demographics to also consider suggesting a default name for the section in which this information should be placed.

Ethical considerations and legal consideration

Above the ethical issues in collecting and reporting demographic information have already been touched upon. All demographic information, but some especially (such as on ethnic background, SES, and disabilities) concern sensitive data that might not be easy to obtain due to objections from parents or schools. Further, if obtained, these data should be handled and reported carefully without the possibility of making the subjects of a study traceable. In behavioral sciences, the collection and report of demographic information of its’ subjects, which also often concerns children, is standard and required. This can be helpful in identifying safe procedures for this type of data (for instance, using careful informed consent procedures and pseudonymisation of the obtained data). Furthermore, the example of behavioral sciences can also clarify the purpose of reporting demographic information that is in itself not part of the research question. The discussion on the representativeness of study samples exists in psychology as well, and one recommendation here is also to consistently report demographics in empirical studies (Rad, Martingano & Ginges, 2018).

The collection and reporting of detailed demographic data could also be hindered by legal concerns about the privacy of the study participants. This is an increasingly important issue, because interest in privacy and personal data protection has intensified, leading to the adoption of new laws, in particular in the European Union. Data sanitation techniques, including randomisation and generalisation, can be applied, aiming to prevent the re-identification of data subjects. The application of these techniques when reporting on demographic data on study participants is challenging, because the more detailed the reported data, the more identifiable the subjects become. For researchers reporting this data, it could understandably be a challenge to achieve a proper balance between on the one hand providing replicable and representative data and on the other hand protecting the privacy of the participants and complying with privacy laws. Regulations and customs from behavioral sciences could be helpful here.

Limitations of current research and directions for future work

This literature review focused specifically on proceedings of the SIGCSE, ITiCSE and ICER conferences. We believe these venues reflect the global CSEd community and provide the valid starting point to explore the demographics of it’s studies’ subjects. At the same time, there is some limitation because of this focus. Future work can look further into papers from CSEd journals, as well as focus on the specific case of local conferences. Moreover, for our analyses we assumed that each of the 143 studies has been reported on in exactly one paper in the dataset; a publication of a study in more than one papers in the three conferences would be a threat to the validity of our findings. Further, the analyses of specific demographic factors in the current research are somewhat limited, not looking for example at relations between different factors. Our goal was to provide an overview of different factors, future studies could focus on specific factors such as ethnic background to gain further insight. Finally, a threat to validity in the method of analyzing the papers should be mentioned. This was done manually by nine different researchers, without applying a method of double-coding. However, we attempted to reduce this threat by first practicing the analysis process with all researchers together and agreeing on common coding conventions, and second by regularly meeting and discussing the general experiences as well as specific doubts about the reported demographic information and especially about information that was not explicitly stated but deduced from the study context. In addition, we make the resulting dataset available to the research community for cross validating our findings.

Concluding remarks

With this paper, we have provided insight in the representativeness of participants in K-12 CSEd studies. Our literature review showed that certain populations (children from the US, boys, and children without previous computer science experience) are over-represented. Further, we showed that certain demographic information is not always or not consistently reported on. This is especially the case for race/ethnic background, SES, and disabilities. We are very much aware of practical, ethical, and legal challenges in collecting demographic information. However, interpretation, generalization, and replication of research findings as well as policy development could benefit greatly from, as much as possible, consistent report of characteristics of populations included. Consequently, an important recommendation would be to develop standards for the collection and report of demographic information in empirical studies in K-12 CSEd. Such guidelines, inspired by behavioral sciences, could include alternative options such as the report at school or neighborhood level, to be applied where appropriate.

Supplemental Information

Supplemental Information 1 Overview of papers included in the 2 phases of the literature review and raw demographics of the papers included in the analysis

Click here for additional data file.

Additional Information and Declarations

Competing Interests

Author Contributions

Data Availability

The authors declare there are no competing interests.

Anna van der Meulen and Alaaeddin Swidan conceived and designed the experiments, performed the experiments, analyzed the data, prepared figures and/or tables, authored or reviewed drafts of the paper, and approved the final draft.

Felienne Hermans conceived and designed the experiments, performed the experiments, analyzed the data, performed the computation work, prepared figures and/or tables, authored or reviewed drafts of the paper, and approved the final draft.

Efthimia Aivaloglou, Marlies Aldewereld, Bart Heemskerk, Marileen Smit, Charlotte Thepass and Shirley de Wit conceived and designed the experiments, performed the experiments, analyzed the data, authored or reviewed drafts of the paper, and approved the final draft.

The following information was supplied regarding data availability:

The raw data are available in the Supplemental File.

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
