# Peer review of "Who participates in computer science education studies? A literature review on K-12 subjects"

_PeerJ Computer Science, doi:10.7717/peerj-cs.807_

## Round 0.1 · original submission · Major Revisions

The reviewers had some positive comments and some areas for improvement. Please review their comments carefully and address them. I look forward to a resubmission.

Reviewer 1 ·

Basic reporting

Is the review of broad and cross-disciplinary interest and within the scope of the journal?

The authors have focused their literature review on three flagship conference proceedings in order to highlight the ways in which researchers describe, or not, the demographics of the students who participated in the studies. The authors maintain that this is important to study for two reasons: first, to inform policy and support replications, and second, to identify demographic attributes that are important for the field to report. As such, the literature review holds broad appeal and ought to be of interest to those working in the multidisciplinary computer science education field.

Has the field been reviewed recently? If so, is there a good reason for this review (different point of view, accessible to a different audience, etc.)?

The authors note that similar review work has been completed (McGill et al., 2018; Schlesinger et al., 2017), although they seem to indicate that their review is distinct in that it evaluates additional demographic information, disability status in particular. I am not certain that the authors have convinced me that this sensitive information can be widely and appropriately asked across K-12 studies, at least those that occur in the United States.

Does the Introduction adequately introduce the subject and make it clear who the audience is/what the motivation is?

The Introduction is well written, situates the review among related research, then appropriately narrows its focus on a brief overview of this study’s process and purpose.

Experimental design

Is the Survey Methodology consistent with a comprehensive, unbiased coverage of the subject? If not, what is missing?

The authors do a commendable job of detailing their process and justifying their use of the three conference proceedings. They do note in Limitations that there was no attempt at inter-rater reliability, or double-coding. This does seem potentially problematic as the authors indicated they deduced information context, rather than from details overtly stated in the studies. It may be that one researcher missed vital information or misunderstood. Such double checking is important to the validity of the authors’ claims.

Additionally, the authors do not address what safeguards were in place to ensure that an intervention was not submitted to and accepted at more than one of these conferences and therefore those data were considered more than once.

Are sources adequately cited? Quoted or paraphrased as appropriate?

The authors cite appropriately, although I am used to alphabetizing in-text citations when there is more than one cite. Relatedly, there are several places (lines 61, 69, and 74) where the formatting of the citation is incorrect.

Is the review organized logically into coherent paragraphs/subsections?

The review is organized well and logically flows from Introduction to Conclusion.

Validity of the findings

Is there a well developed and supported argument that meets the goals set out in the Introduction?

I provide more specific information below in General Comments, but overall I am struggling to see the feasibility of including disability status as a socio-demographic metric. I understand the authors’ contention that such a characteristic needs to be considered for generalizability purposes; however, the reality of receiving such information, at least in the US, is highly complicated and would not lend itself to generalizability. This aside, the authors complete their literature review as they set out in the Introduction.

I am not certain that the statement “Both for race/ethnic background and for disabilities a pattern can be seen where these categories are only reported on when deviating from the majority” is supported. Is it that these characteristics are only reported when they differ from the majority or could it be that these socio-demographic metrics are problematic to collect?

Does the Conclusion identify unresolved questions / gaps / future directions?

The Conclusion identifies suggestions for making consistent across CSEd studies the ways we report on demographics. What the authors need to include are real ways to balance the tension between sharing empirical findings about specific students in K-12 studies when the data we are able to collect might be at the school level only.

I feel the authors could strengthen their argument and the paper overall by delving more deeply into the articles that report on disability status, as this seems to be their major interest, especially those not focused solely on a certain population of students (ie., blind students). How did these articles report on these disabilities? What was their institutional review policy? How did they obtain this information (ie., from the students, teachers, or parents)? What major findings/conclusions did they offer? The number of articles that fall into this category is quite small, so more closely exploring them may help draw out and support your contention that disability status is a metric essential to report. Given this small number, perhaps it would be wise to broaden your search beyond the flagship conferences. In other words, the authors could share their literature review findings, including the small number of articles that report on disability status, and then provide a detailed overview of articles in CSEd at large (conferences and journals) that do report on disability status.

Additional comments

The following are line item/section comments regarding grammar, formatting, or general structure:

Line 5 (Abstract): if space permits, it may be helpful to include the year range
Lines 25, 38, and 50: authors appear to use ‘pre-college,’ ‘pre-university,’ and ‘K-12’ interchangeably. I think it would read better if one phrase were used; K-12 is most common. These phrases appear throughout the article.
Line 99: it seems the RQ should end as “... in K-12 programming” and not into
Line 156: grade 6 is included twice in both elementary and middle school; where did you ultimately include it?

Gender: is it important to include the gender options the studies permitted the participants to select, or that --% of studies permitted students to opt out or select Other? Also, the finding that males are overrepresented is not terribly surprising, but I feel this interacts with the context of the studies. For example, did the studies where males were overrepresented take place in elective classes or optional experiences such as camps?

Race/Ethnic Background
Line 178: the sentence that begins on line 177 and includes “studies is represented in,” is poorly worded or missing a reference to a figure.
“This is further confirmed by the large number of studies that report on mixed-racial experimental groups, as Figure 3 depicts, which does not correspond with how mixed classrooms typically are in the US (Stroub and Richards, 2013)”: what does this mean? What are mixed classrooms and how are they typically organized/studied in the US?

SES and Disabilities
I have combined these two subsections as my concerns with them are similar.
As a US researcher, I have always used NSLP (free and reduced school lunch eligibility) as a proxy for SES. This follows because we have never found it appropriate to ask students about their family’s SES and this information is publicly available at the school level. When working with a single class or a subset of students in class, it feels inappropriate to report out the school level SES.

No university IRB would approve US researchers asking K-12 students, deemed a vulnerable population, about their disability status. FERPA regulations prohibit school personnel from disclosing such information at an individual level to outside researchers. It may be that a teacher could tell a researcher something generic, such as “of the 30 students in this classroom, 5 have IEPs and 4 have 504 plans.”

Line 216: “However, given the aforementioned prevalence of individuals with a disability, it seems unlikely that the participant groups of the other 128 papers did not include any of these children”: of course these classrooms included students with disabilities, but the practice of mainstreaming and not delineating students by ability, likely contributed to the small number of studies that report on this characteristic

Lines 273-4: the wording of the final sentence is problematic. You have set up a juxtaposition whereby poor inner city cannot include well-educated. Don’t conflate education and income here. Something like “it could be a school in a poor inner city area or a high-income neighborhood” would be better.

·

Basic reporting

In the introduction, the authors adequately introduce the topic by providing sufficient background on pre-university programming education and comprehensively discussing related work on demographics in computer science education studies. They clearly distinguish their work from prior research (e.g., by McGill et al. (2018)) and identify gaps. The desideratum on which the authors build is that there is no comprehensive research on the demographics of computer science education studies between 2014 and 2018.

Overall, the paper is well structured and contains detailed sections on methodology, results, and conclusions. The argumentation is comprehensive throughout and easy to follow. Relevant previous literature is referenced appropriately. Figures and tables are appropriately labelled and well described in the respective sections/paragraphs.

The paper is written in professional, unambiguous English. The text is technically correct and follows professional standards of courtesy and expression.

Suggestions for improvement:

- In the “Pre-university Programming Education” section, two improvements should be made to the citation style: (1) (Yang et al., 2015) for example […] -> Yang et al. (2015) for example; (2) Aivaloglou and Hermans (Aivaloglou and Hermans, 2016) performed an […] -> Aivaloglou and Hermans (2016) performed an.

- Authors should consider submitting the figures in a better resolution as the diagrams provided are slightly blurred. Please, add a dot after each of the figure labels.

- Please, reference Figure 2 in the text. Currently, there is no reference to it in the text.

- Please, use CSed abbreviation consequently.

Experimental design

The paper is a literature review with a clear formulated study design. It aims to address the wide computer science education community which is in line with the aims and scope of the PeerJ journal.

In the methodology section, the authors clearly articulate the research question (What are the demographics of subjects who participate in CSEd studies into K-12 programming?) and comprehensively describe the procedure for literature selection, narrowing the scope of the selected literature to studies of the pre-university target group with the focus on programming education. The selection of demographic categories is clearly reasoned, and the description of the analysis process is extensively described. The paper also includes a detailed discussion of the limitations of the literature review. All raw data is provided for better understanding and comprehension of the results, which I found helpful.

Suggestions for improvement:

- Although the authors refer to computer science education in the title of the paper, the literature used for the review is limited to programming education only. However, computer science education is much broader than learning programming. Please, include this information as a limitation and explain in the introduction why you focused on programming education as one of the aspects of computer science education.

- In the introduction, the authors justify the selection of SIGCSE, ITiCSE and ICER by arguing that these are “flagship conferences” in the CSEd community. Please, support this argument with references.

- In presenting the results in the “Race/Ethnic Background” section, the authors go beyond describing the results (“This leads us to hypothesize that reporting bias is at play here”). However, since this argument is already about evaluating the results, I recommend moving it to the “Ethical and Legal Considerations” section.

Validity of the findings

The findings are appropriately structured and presented according to the categories proposed in the survey methodology section. In the conclusions, the authors reflect on the findings based on the research question and the state of the arts presented in the introduction. The authors also identify future directions relevant to the entire computer science education community, suggesting, for example, the development of standards/guidelines for collecting and reporting demographic information in empirical studies of K-12 CSed.

Overall, I found the conclusions to be well articulated and linked to the original research question. The authors fully achieved the objectives stated in the introduction and survey methodology.

Additional comments

I find the results of this literature review enriching for research in computer science education. Many thanks to the authors of this work!

---

## Round 0.2 · accepted · Accept

Thank you for addressing the reviewers' previous concerns. This manuscript will be a helpful contribution to the literature on this topic.

Reviewer 1 ·

Basic reporting

The authors have addressed all my previous concerns.

Experimental design

The authors have addressed all my previous concerns.

Validity of the findings

The authors have addressed all my previous concerns.

·

Basic reporting

In the introduction, the authors adequately introduce the topic by providing sufficient background on pre-university programming education and comprehensively discussing related work on demographics in computer science education studies. They clearly distinguish their work from prior research (e.g., by McGill et al. (2018)) and identify gaps. The desideratum on which the authors build is that there is no comprehensive research on the demographics of computer science education studies between 2014 and 2018.

Overall, the paper is well structured and contains detailed sections on methodology, results, and conclusions. The argumentation is comprehensive throughout and easy to follow. Relevant previous literature is referenced appropriately. Figures and tables are appropriately labelled and well described in the respective sections/paragraphs.

The paper is written in professional, unambiguous English. The text is technically correct and follows professional standards of courtesy and expression.

The authors included suggestions for improvement from the first review.

Experimental design

The paper is a literature review with a clear formulated study design. It aims to address the wide computer science education community which is in line with the aims and scope of the PeerJ journal.

In the methodology section, the authors clearly articulate the research question (What are the demographics of subjects who participate in CSEd studies into K-12 programming?) and comprehensively describe the procedure for literature selection, narrowing the scope of the selected literature to studies of the pre-university target group with the focus on programming education. The selection of demographic categories is clearly reasoned, and the description of the analysis process is extensively described. The paper also includes a detailed discussion of the limitations of the literature review. All raw data is provided for better understanding and comprehension of the results, which I found helpful.

The authors included suggestions for improvement from the first review.

Validity of the findings

The findings are appropriately structured and presented according to the categories proposed in the survey methodology section. In the conclusions, the authors reflect on the findings based on the research question and the state of the arts presented in the introduction. The authors also identify future directions relevant to the entire computer science education community, suggesting, for example, the development of standards/guidelines for collecting and reporting demographic information in empirical studies of K-12 CSed.

Overall, I found the conclusions to be well articulated and linked to the original research question. The authors fully achieved the objectives stated in the introduction and survey methodology.